# The Role of Prussian Blue-Thallium and Potassium Similarities and Differences in Crystal Structures of Selected Cyanido Complexes of W, Fe and Mo

**DOI:** 10.3390/ma15134586

**Published:** 2022-06-29

**Authors:** Maciej Hodorowicz, Janusz Szklarzewicz, Anna Jurowska

**Affiliations:** 1Faculty of Chemistry, Jagiellonian University, Gronostajowa 2, 30-387 Kraków, Poland; janusz.szklarzewicz@uj.edu.pl (J.S.); jurowska@chemia.uj.edu.pl (A.J.); 2Institute of Chemistry, Jan Kochanowski University in Kielce, 7 Uniwersytecka Str., 25-406 Kielce, Poland

**Keywords:** cyanides, iron, molybdenum, structure, thallium, tungsten

## Abstract

The synthesis and single X-ray crystal structures of Tl_2_[W(CN)_6_(bpy)]·H_2_O, Tl[W(CN)_6_(bpy)], Tl_57_[{Fe(CN)_6_}_12_{NO_3_}_9_]·9H_2_O, Tl_4_[W(CN)_8_] and Tl_4_[Mo(CN)_8_] are described. Salts crystalize in the monoclinic or trigonal (Fe) space group. For all complexes, the very unusual “side-on” coordination of Tl to C and N of cyanido ligands is observed and compared to potassium cations. The very strong bonding to cyanido ligands with an 8–13 coordination number of thallium is described and discussed. The difference between potassium and thallium bonding is an indication of the biological activity of thallium and the role of Prussian blue in the thallium detoxification procedure.

## 1. Introduction

Thallium poisoning results from the similarity between thallium and potassium ions. They have similar ionic radii, and experimental results indicate that thallium ions exchange potassium ions in numerous metabolic pathways. It has been found that in frog muscle, potassium and thallium ions traverse cell membranes in a similar manner [1]. An active potassium transport mechanism in rabbit erythrocytes has also been shown to transport thallium [2], and subsequently the same authors described a relationship between potassium levels and thallium excretion in rats, dogs, and sheep [3]. The diagnosis of thallium poisoning is difficult because it is often unexpected and therefore unfortunately too late. Additionally, the symptoms of poisoning with this element often resemble those of more common neurological diseases. Since 1971, there have been papers reporting the use of Prussian blue, (Fe^III^)_4_[Fe^II^(CN)_6_]_3_, as a fairly effective agent in the treatment of thallium ion poisoning [4]. This compound is administered orally; it adsorbs thallium ions and exchanges them for potassium ions. Unfortunately, the almost insoluble Prussian blue, used in thallium poisoning, penetrates the cell membranes poorly (in water, it forms a suspension) and binds slowly to thallium ions. This is a result of the polymeric nature of this compound and the compact packing of the structure without channels for ion penetration.

Hexacyanoferrates exhibit a wide variety of compositions and structures, not only dependent on the cation used but also on the method of preparation [5,6,7,8]. For the same transition metal, different compositions can be obtained and different structural arrangements can be observed, which is due to the fact that the nature of the preparation process strongly depends on the composition and crystal structure of the starting solid [9,10]. Because of the toxicity of thallium and its ions and the resulting reluctance to work with this element, the chemical or coordination properties are little understood, although in many ways unique [11]. This is due, among other things, to the fact that the thallium ion, Tl(I), is a monovalent metal ion in the p block of the periodic table, whose configuration contains a closed s subshell. The valence shell electron configuration of d^10^s^2^ with a lone pair results in weak covalent interactions, and the Tl(I) compounds tend to exhibit ionic interactions. Additionally, the structural chemistry of thallium compounds shows unique features due to its ability to form Tl-Tl and Tl-C bonds, the large radius of thallium ions and the high coordination number, and the presence of an electron pair in the valence shell [12,13,14,15,16,17,18,19,20,21]. The variety of structures formed by hexacyanoferrates(II, III) and the different binding modes of thallium result in difficulty of explanation of how thallium binds to Prussian blue. All this has resulted in the role of Prussian blue and the method of thallium removal being a matter of speculation rather than fully understood.

In this respect, the 6-cyanide anion [W(CN)_6_(bpy)]^2−^ completely contrasts with the [Fe(CN)_6_]^4−^ complexes. In our previous studies, we have shown that, contrary to hexacyanoferrates, in the case of salts of [W(CN)_6_(bpy)]^2−/−^, the anion structure is very stable; its deformations are small, and they do not depend on the size of the cations (their radius) and the type of intermolecular interactions [22,23,24,25,26,27]. This stability of the geometry of the anionic structure and its negligible susceptibility to the packing effect is a very important and valuable feature from the point of view of synthesis and structures based on heavy-metal and transition-metal cations, in which simple electrostatic anion–cation interactions do not occur, but polymers with cyanide bridges are formed. We showed earlier that potassium, rubidium, and caesium cations, contrary to hexacyanoferrates(II, III) and octacyanido complexes of Mo(IV) and W(IV), do not form weak electrostatic cation–anion interactions but bind very strongly to cyanido ligands in the [W(CN)_6_(bpy)]^2−^ anion, forming structures that could be described as covalent coordination polymers. In this sense, systems containing the monomeric, thermodynamically very stable and kinetically very inert anions of [W(CN)_6_(bpy)]^2−/−^, where alkali metal salts are highly soluble not only in water, could be an alternative to Prussian blue. The high affinity to potassium cations with the formation of almost covalent bonds to cyanido ligands, as reported in our previous work, forces us to study the coordination properties of this anion toward thallium anions in the context of their application as potential drugs used for binding and removal of thallium ions from the body. An additional advantage supporting this application of these systems is the fact that the solubility of thallium salts presented in this manuscript is inferior to that of their analogues containing sodium or potassium cations. In the present work, structural studies were carried out to not only to increase our knowledge of the thallium bonding to [W^IV^(CN)_6_(bpy)]^2−/−^ anions but to compare the thallium coordination modes to other cyanido systems, in particular to [Fe^II^(CN)_6_]^4−^, [W^IV^(CN)_8_]^4−^, and [Mo^IV^(CN)_8_]^4−^. In this paper, we present and discuss five new single crystal structures of thallium cyanido complexes, almost 20% of all known structures, as there are only 27 known structures with Tl-NC bonds.

## 2. Materials and Methods

K_2_[W(CN)_6_(bpy)]·3H_2_O, K[W(CN)_6_(bpy)], K_4_[W(CN)_8_]·2H_2_O, and K_4_[Mo(CN)_8_]·2H_2_O were synthesized as we described earlier [26,28,29,30]. All other chemicals were analytical grade (Aldrich) and were used as supplied. Microanalyses of carbon, hydrogen, and nitrogen were performed using an Elementar Vario MICRO Cube elemental analyzer. IR spectra were recorded on a Nicolet iS5 FT-IR spectrophotometer.

### 2.1. Synthetic Procedures

**Synthesis of Tl_2_[W(CN)_6_(bpy)]****⋅****H_2_O (1).** K_2_[W(CN)_6_(bpy)]·3H_2_O (0.100 g, 0.16 mM) in 2 mL of H_2_O was mixed with a saturated solution of TlNO_3_ (0.128 g, 0.48 mM) in 2 mL of H_2_O. The mixture was left for crystallization. The next day, the formed very-dark-red crystals were filtered off, washed with a small amount of cold water, and dried in air. Yield 0.11 g; 74%. Anal. Calcd. for C_16_H_10_N_8_OTl_2_W: C, 20.82; N, 12.14; H, 1.09%. Found: C, 20.41; N, 11.85; H, 1.16%. FT-IR (ATR, cm^−1^): 3553 (m), 3369 (m), 3102 (w), 3076 (w), 2110 (w), 2101 (s), 2085 (vs), 1606 (m), 1477 (m), 1434 (m), 1327 (w), 1285 (w), 1254 (w), 1170 (w), 1078 (w), 1017 (w), 967 (w), 799 (w), 773 (s), 730 (w), 611 (w), 474 (w), 436 (w).

**Synthesis of Tl[W(CN)_6_(bpy)] (2).** K[W(CN)_6_(bpy)] (0.050 g, 93 mM) in 2 mL of water was mixed with a saturated solution of TlNO_3_ (30 mg, 11 mM) in water. The mixture was left for crystallization in the dark. The formed light pink crystals were filtered off, washed with a small amount of cold water, and dried. All manipulations were performed in the dark. Anal. Calcd. for **2**·2H_2_O; C_16_H_12_N_8_O_2_TlW: C, 26.07; N, 15.21; H, 1.63%. Found: C, 25.50; N, 15.20; H, 1.08%. FT-IR (ATR, cm^−1^): 3204 (w), 3131 (m), 3103 (m), 3076 (m), 2141 (w), 1719 (w), 1609 (vs), 1563 (w), 1507 (m), 1476 (m), 1444 (s), 1384 (w), 1326 (s), 1297 (w), 1239 (w), 1223 (w), 1187 (w), 1165 (w), 1130 (w), 1108 (w), 1080 (w), 1042 (w), 1031 (m), 1018 (w), 969 (w), 901 (w), 821 (w), 776 (vs), 726 (m), 660 (w), 641 (w).

**Synthesis of Tl_57_[{Fe(CN)_6_}_12_{NO_3_}_9_]·9H_2_O (3).** K_4_[Fe(CN)_6_]·3H_2_O (0.400 g, 0.95 mM) in 3 mL of water was mixed with TlNO_3_ (1.77 g, 6.6 mM) in 4 mL of water. The mixture was heated to ca. 60 °C until a transparent solution was obtained. This was left for crystallization. The light yellow crystalline product was filtered off, washed with a small amount of water, and dried in air. Yield 1.10 g; 94%. Anal. Calcd. for C_72_Fe_12_H_2_N_81_O_36_Tl_57_: C, 5.80; N, 7.61; H, 0.013%. Found: C, 5.93; N, 7.60; H, 0.30%. FT-IR (ATR, cm^−1^): 2010 (vs), 1594 (w), 1320 (m), 579 (w).

**Synthesis of Tl_4_[W(CN)_8_] (4).** K_4_[W(CN)_8_]·2H_2_O (0.4 g; 0.69 mM) in 2 mL of water was mixed with TlNO_3_ (0.80 g, 3.00 mM) in 2 mL of water. The mixture was heated until a transparent solution was obtained. This was left aside for crystallization. Formed light orange crystals were filtered off, washed with a small amount of water, and dried in air. Yield 0.75 g, 91%. Anal. Calcd. for C_8_N_8_Tl_4_W: C, 7.94; N, 9.26%. Found: C, 7.88; N, 9.18%. FT-IR (ATR, cm^−1^): 2098 (m), 2088 (w), 2072 (s).

**Synthesis of Tl_4_[Mo(CN)_8_] (5).** K_4_[Mo(CN)_8_]·2H_2_O (0.4 g; 0.81 mM) in 2 mL of water was mixed with TlNO_3_ (0.90 g, 3.38 mM) in 4 mL of water. The mixture was heated until a transparent solution was obtained. This was left aside for crystallization. Formed light orange crystals were filtered off, washed with a small amount of water, and dried in air. Yield 0.84 g, 93%. Anal. Calcd. for C_8_MoN_8_Tl_4_: C, 8.56; N, 9.99%. Found: C, 8.52; N, 9.92%. FT-IR (ATR, cm^−1^): 2100 (m), 2091 (w), 2077 (s).

### 2.2. Crystallography

**Crystallographic data collection and structure refinement.** Crystals suitable for X-ray measurements were taken from solution (prepared as described in experimental section) directly before filtration. Diffraction intensity data for single crystals of five new compounds were collected on the Rigaku XtaLAB Synergy-S diffractometer with mirror-monochromated Mo Kα and CuKα radiation for **1**, **3**, **4**, **5**, and **2**. Diffraction measurements were performed at temperatures: for **1** at 281 K, for **2** at 130 K, while for **3**, **4**, and **5** at 293 K. Cell refinement and data reduction were performed using firmware [31]. The positions of all atoms other than hydrogen were determined by the heavy atom method using SIR2014 (structures **1** and **2**) or SIR2019 (structures **3**, **4**, and **5**) [32]. All non-hydrogen atoms were refined anisotropically using weighted full-matrix least-squares on *F*^2^. Refinements and additional calculations were carried out using SHELXL-2017/1 (**1**) or 2019/2 (other compounds) [33]. All hydrogen atoms joined to carbon atoms were positioned with idealized geometries and refined using a riding model with *U*_iso_(H) fixed at 1.2 *U*_eq_(C_arom_). The figures were made using Diamond ver. 4.6.7 software [34]. CCDC 1854226, 2158318, 2160838, 2160846, and 2161288 contain the supplementary crystallographic data for **1**, **2**, **3**, **4**, and **5**, respectively. These data can be obtained free of charge from The Cambridge Crystallographic Data Centre via www.ccdc.cam.ac.uk/data_request/cif, accessed on 6 July 2018.

## 3. Results

The thallium cation is a unique cation that resembles alkali metal cations; the thallium radius is almost identical to that of K^+^. This is why thallium cations can exchange potassium ones in biological systems and are responsible for their high toxicity. Thus, it was interesting for us to study the structures of cyanido complexes with thallium, especially because, as mentioned in the introduction, hexacyanido iron complexes are used in medical treatment in thallium poisoning. The reaction of selected cyanides of Fe(II), Mo(IV), W(IV), and W(V) with TlNO_3_ in aqueous solution results in the isolation of well-shaped crystals of salts **1**–**5**. All salts are much less soluble than their potassium analogues; however, they remain well soluble in water, DMF, and DMSO, sparingly soluble in polar solvents (such as MeOH, EtOH, and MeCN), and almost insoluble in nonpolar solvents. The IR spectra (Appendix A) are consistent with the salt composition; the most interesting is the strong changes in the ν_CN_ region compared to substrates in K^+^ form. This indicates strong interactions of cyanido ligands with thallium cations; for salt **3**, it was also impossible to remove NO_3_^−^ anions by recrystallization, suggesting its important role in crystal formation. All this is explained in the crystal structure description.

### 3.1. Crystallographic Studies

The crystal data and structure refinement parameters for complexes **1**–**5** are shown in Appendix A. Selected bond and angle parameters are shown in Appendix A. All complexes studied gave well-shaped single crystals whose colour was of anion origin. Crystals were stable in air and were taken for X-ray measurements directly from the mother liquor, as described previously. All salts (Appendix A) crystallized in the monoclinic space group, except **3,** in which a rare trigonal group was found.

#### 3.1.1. Structures of [W^IV^(CN)_6_(bpy)]^2−^ (**1**) and [W^V^(CN)_6_(bpy)]^−^ (**2**) Anions

In complex **1**, as well as in other previously published compounds [22,23,24,25,26,27], the anionic group [W(CN)_6_(bpy)]^2−^ adopted a square antiprism geometry, and the coordination environment of the tungsten W(IV) atom consisted of a C_6_N_2_ system with a bpy substituent and two cyanide ligands in one polyhedral square plane and four other cyanido ligands in the other (Figure 1a).

In case **1**, the W-N_bpy_ bond lengths are 2.227(5) and 2.211(5) Å and are almost identical to those observed in the other structures and fall within the range. The values of these bonds are almost identical to those in (PPh_4_)_2_[W(CN)_6_(bpy)] (2.220 Å) [35]. The average W-C_cyanido_ bond lengths in the anion at **1** and **2**, 2.158 Å and 2.165 Å, respectively, have similar values relative to the bond lengths in the octacyanido anions [W(CN)_8_]^4−^ (2.155 Å), [Mo(CN)_8_]^4−^ (2.164 Å) [29,36] and also in the potassium salts of [W(CN)_6_(bpy)]^2−^ (2.159 Å) and [W(CN)_6_(bpy)]^−^ (2.156 Å) [26,28]. It is also worth noting that the distances between W-N_cyanido_, in all the described compounds, are very close to those found for K_4_[W(CN)_8_], where the average length of the W-N_cyanido_ bond is 3.30 Å [36], indicating a strong influence of the bpy ligand only on the first coordination sphere of the W(IV) and W(V) atoms, while the second one remains unchanged. In **1** and **2**, incidentally, as in already published alkali metal salt research [23,24,25,26], [W(CN)_6_(bpy)]^2−/−^ anion–cation interactions differ in character from those observed in the tetraphenylarsonium salt [23].

The presence of hydrophobic organic substituents in the cations precludes the occurrence of ionic interactions observed in alkali metal salts, but the geometry of the coordination environment in [W(CN)_6_(bpy)]^2−^ for systems containing AsPh_4_^+^ and PPh_4_^+^ cations is very similar to that observed for alkali metal salts. This observation confirms the stiffening effect of the bpy ligand and thus shows that the coordination geometry of [W(CN)_6(_bpy)]^2−^ anions is little affected by intramolecular interactions, which is an advantage over other octacoordination complexes, in particular Mo and W, in which intermolecular interactions very often reduce the symmetry of the anion from D_4d_ to D_2d_ or even lower [37]. Such low susceptibility to anion geometry changes depending on the structure packing scheme is important for the reproducibility of the synthesis and the possibility of structure prediction, in particular for systems containing transition metal cations and cyanidometalate anions, in which an important structural role is played not only by simple electrostatic interactions of the cation–anion type but also by cyanide bridges, which are responsible for the formation of inorganic polymeric structures. The anionic complex [W(CN)_6_(bpy)]^−^ in structure **2** has a dodecahedral geometry, as shown in Figure 1b.

The ligand C2-N2 (perpendicular to the plane of the bpy ligand) of the cyanide is slightly inclined toward the bpy ligand, while the other cyanido ligands face outward from the bpy ligand. An analogous geometrical fact has been observed for already published structures [28]. This type of slope of the single cyanido ligand to the plane of the bpy ligand is also observed for structure **1** and previously published compounds containing [W(CN)_6_(bpy)]^2−^ [22,23,24,25,26], but the slope is 3–4° lower. The bipyridyl ligand (bpy) present in structures **1** and **2**, which is the direct coordination environment of the tungsten atom in [W(CN)_6_(bpy)]^2−/−^, has an almost completely flat molecular geometry.

On the basis of the structural analysis of compounds **1** and **2**, it can be concluded that the observations noted in the already published structures of +1 cation salts are confirmed. Thus, it can be stated that the average W-C distance (2.155 Å) in compounds in which tungsten is on the fifth oxidation state, W(V), is almost identical to that of the W(IV) analogues (2.153 Å) [22,23,24,25,26]. Thus, on the basis of changes in the length of the W-C and W-N bonds, it is not possible to unambiguously distinguish the oxidation state of tungsten in the studied anions. However, in the analogues with W(V) [28] in complex **2**, the valence angle C15-W1-C18 has a larger value (121.19°) compared to the corresponding angle in the group of compounds with W(IV), for which the angle is 116.61°. This is a regularity observed in several structures, so one can think that this angle can be a very good indicator of the degree of tungsten oxidation in the described systems. Another important structural difference between the anionic complexes [W^IV^(CN)_6_(bpy)]^2−^ and [W^V^(CN)_6_(bpy)]^−^, (**1**) and (**2**), is that the C14-W-C16 angle (101.64°) for **2** has a much smaller value compared to the corresponding angle in the W(IV) analogues, which is 108.18° for **1**.

#### 3.1.2. Structure **1**

The structure and content of the asymmetric part of the Tl_2_[W(CN)_6_(bpy)]⋅H_2_O (**1**) unit cell with the atomic labelling scheme is shown in Figure 1a. It is composed of an anionic part containing [W(CN)_6_(bpy)]^2−^ and a cationic part containing two thallium(I) ions, and in addition the chemical composition is completed by one water molecule. Each [W(CN)_6_(bpy)]^2−^ anion of compound **1**, through bridging CN^−^ ligands, interacts with as many as six Tl(I) cations (Tl1, Tl2 and their symmetrical counterparts Tl1′ [x, y, −1 + z], Tl1″ [1/2 + x, 1/2 − y, −1/2 + z], Tl2″ [1/2 + x, 1/2 − y, −1/2 + z], Tl2″ [−1/2 + x, 1/2 − y, −1/2 + z]), through “end-on” and “side-on” cyanide W-CN-Tl bridges.

The Tl1 thallium cation is “side-on” manner coordinated by the three cyanido ligands of the anion [W(CN)_6_(bpy)]^2−^ (Figure 2), and the Tl-N and Tl-C distances of this interaction are on average 3.000 and 3.469 Å, respectively, with the mean C-N-Tl angle being 104.46°.

In addition, the Tl1 ion is coordinated by one (O1) water molecule as well as one N atom (N5), which typically interact with the thallium cation to form an “end-on” interaction. The Tl2 cation is coordinated analogously, except that all interactions through the cyanido ligands are “side-on”. It should be mentioned that almost identical interactions were observed for structures containing potassium cations, with the difference that thallium ions are located closer to the tungsten atom (stronger cyanide bridges) [26], which on one hand explains much lower solubility of Tl_2_[W(CN)_6_(bpy)] relative to K_2_[W(CN)_6_(bpy)] and on the other hand confirms high chemical similarity of K^+^ and Tl^+^ ions. Note that by sharing two cyanido ligands, C1-N1 and C3-N3, and a water molecule present in the structure, the thallium cations form a dimeric system. The Tl1–Tl2 distance of 4.125 Å is not much larger than the sum of the van der Waals radii (1.96 Å by [38] and 1.98 Å by [39]). In addition, based on the literature data, it is difficult to conclude that there are any direct Tl-Tl type interactions between the thallium ions described in the literature [17,18]. The coordination environments around Tl1 and Tl2 adopt the geometry of a strongly deformed trigonal prism.

As a result of the coordination of the cyanido ligands and their bridging action, as well as the bridging role of the water molecule O1 to bind the cations Tl1 and Tl2 together, a layered two-dimensional network (010) is formed that extends along the crystallographic axes of [100] and [001] (Figure 3, left).

The bpy ligands of the neighbouring anions [W(CN)_6_(bpy)]^2−^ partially overlap, such that each bpy ligand is arranged in a π-stack with its neighbours, with a separation of 3.993(4) Å between parallel overlapping ring groups. Extended weak π…π interactions (Table 1) are responsible for linking the individual layers into a coherent 3D structure (Figure 4). Additionally, the network of hydrogen interactions present in the structure stabilizes the inorganic layer (Table 2). The shortest interlayer (between adjacent inorganic layers A) W1-W1 [2 − x, −y, 1 − z] distance is 7.938(7) Å, while the shortest intralayer (within inorganic layer A) W-W distances are: vertical W1-W1 [1/2 + x, 1/2 − y, 1/2 + z] = 7.920(6) Å and horizontal W1-W1 [1 + x, y, z] = 8.572(4) Å. Comparing the packing of structure **1** to its potassium counterpart [26], one can see that the inorganic layer does not have as clearly distinguished a tungsten and thallium sublayer as it does in the case of the potassium cation derivative. This indicates that the thallium ions are more strongly bound to the CN- ligands, making compound **1** better crystallized than its potassium counterpart and thus less soluble.

#### 3.1.3. Structure **2**

The asymmetric part of the elemental cell of structure **2** consists of a [W(CN)_6_(bpy)]^−^ anion and a thallium cation (Figure 1b), Tl1, which is coordinated by five cyanido ligands, two of which are “side-on” and the other three “end-on” (Figure 5).

“Side-on” interactions between cyanides (π electrons of the cyanide group) and alkali metal ions were observed previously by Rauchfuss et al. in caged cyanometalanes [40,41,42]. Similar interactions were also described in the work of Gokel et al., except that the “side-on” interactions involved alkene/alkene units and alkali metal cations [43,44,45,46]. In addition, in our earlier work on monocationic systems (K^+^, Rb^+^, Cs^+^) with [W(CN)_6_(bpy)]^2−/−^, “side-on” cyanide interactions were described, which were obviously different from the cyanide interactions described here [23,26].

Identical coordination modes were observed for analogous potassium structures for both W(IV) and W(V) analogues, although in this case their ratio was reversed [22,23,24,25,26,27,28]. Interestingly, the coordination number of the thallium cation is 5 and is identical to the W(IV) analogue **1**. It was completely different for the potassium structures, because in the W(IV) analogue the coordination of the potassium cation was higher, and in the case of K1 it was 8 and even 11 if K-C and K-N are treated as separate bonds. In the same structure, the other potassium cation K2 showed a coordination of 6 [26]. In the potassium analog containing W(V), the salt was anhydrous [28], and the potassium atom was coordinated to only six cyanide ligands. For structure **2**, the situation was identical, which may indicate that in systems with W(V), the Tl(or K)-N bonds are stronger than those of Tl(or K)-H_2_O, and water molecules are removed from the structure during the crystallization process. The symmetry of the coordination environment around the thallium cations in structure **2**, analogous to **1**, is ambiguous, but it can be considered as a distorted square pyramid in which the values of the N-W-N valence angles deviate from the theoretical ones. The structures of all previously published systems containing alkali metal cations with the anionic complex [W^IV^(CN)_6_(bpy)]^2−^ were characterized by a layered architecture consisting of an alternation of two layers, an inorganic one (composed of cations and anions) and an organic one, formed by planar bpy ligands interacting with each other through weak π…π-type intermolecular interactions [22,23,24,25,26,27]. In the case of structure **2**, however, no such layering was observed, but a three-dimensional spatial arrangement.

From Figure 3 (right), it can be seen that the bpy ligands form a separate layer, but unlike in the W(IV) salt, the W-CN cationic bridging bond is parallel to each bpy molecule. Therefore, it can be concluded that in structure **2** the dominant role in the stabilization of the structure is played by strong anion–cation interactions and not the π…π interactions. Another difference between the complexes **1** and **2** is the fact that in **1** the C2-N2 cyanido ligand, located perpendicular to the plane of the bpy ligand, does not participate in bridging or intermolecular interactions with cations contrary to salt **2**. Moreover, in the structures of the W(IV) analogues, “free” cyanido ligands (not participating in the formation of any interactions) have always been observed, while in the W(V) systems all cyanido ligands are bound to cation atoms.

#### 3.1.4. Structure **3**

Compound **3**, Tl_57_[{Fe(CN)_6_}_12_{NO_3_}_9_]·9H_2_O, crystallizes in the trigonal *R*-3*c* space group (in the rhombohedral setting) with *Z* = 2 molecules per unit cell. The structure consists of four Tl(I) cations, two [Fe^II^(CN)_6_]^2−^ and one NO_3_^−^ anion and a crystallization water molecule (Figure 6).

There are two independent iron atoms in the asymmetric unit, Fe1 in position (2b) on a 3¯ axis and Fe2 in position (6e) on a 2-fold axis. Both Fe atoms are coordinated by the C atom of the cyanido ligand [d(C–N) = 1.153(3) Å and Fe–C–N angle of 179.6(2)°], and their cyanide coordinate environment adopts the geometry of the nearly perfect octahedron. Independent Fe2–C distances are 1.909(11), 1.910(11), and 1.885(11) Å with Fe1–C 1.922(9) Å distances, which are equivalent to those found in other hexacyanoferrates [47,48,49,50,51], thus confirming the presence of Fe(II). The coordination geometry of the four thallium ions is listed in Appendix A and shows interactions with several nitrogen atoms of the [Fe^II^(CN)_6_]^4−^ units and the oxygen atoms of the nitrate ion and water molecule. Due to its position in the special position, on the 3¯ axis, the thallium cation Tl1 has an occupancy of 1/6 (0.1666), and there are only six of this type of cation in the unit cell, in contrast to the cations Tl2, Tl3, and Tl4, all in the general positions, of which there are 36 per unit cell. In addition, the Tl1 cation is the only one bound to six cyanido ligands, forming “side-on” bridges (interaction with the π electrons of the cyanide group) as indicated in Figure 7. The cyanido nitrogen atoms are at a distance of 3.030 Ǻ, while the carbon atoms are at a distance of 3.355 Å. The coordination environment of this cation adopts an octahedral geometry. Tl2 is coordinated by four cyanido ligands, of which two cyanido bridges are “end-on” (2.775 Å N1C1, 2.830 Å N3C3) and the other two bridges are “side-on” (N2C2, N4C4), with distances Tl-N = 3.025–3.038 Å and Tl-C = 3.418–3.511 Å. There are only two atoms in the immediate coordination neighbourhood of the Tl3 cation: the oxygen atom, O1, of the nitrate group being at a distance of 2.860 Å, and the cyanido nitrogen atom, N2 (2.833 Å). In the case of Tl4, the coordination environment is formed by two oxygen atoms: the first, O2, being the oxygen of the nitrate group, and the second, O3, being the water molecule. Additionally, the mentioned thallium cation is coordinated by a cyanido nitrogen atom N4 (Figure 7).

There are as many as eight thallium cations in the immediate vicinity of the nitrate anion (Figure 8a), with the Tl-O distances in the 2.860–3.289 Å range; however, considering the sum of the lengths of the ionic radii of thallium and oxygen, Tl^+^ = 1.50 Å and O^2−^ = 1.26 Å [52], only four thallium cations can be considered as strongly interacting with one NO_3_^−^ molecule (Figure 8b). Each oxygen atom in the nitrate interacts with thallium ions, except that the O1 atom interacts with two Tl3 cations, while the other two oxygen atoms interact with only one Tl4 cation. Due to the network of interactions formed, critical to obtaining compound **3** is the presence of a nitrate anion, which stabilizes the structure in terms of charge compensation. The [Fe^II^(CN)_6_]^4−^ and nitrate anions form a global anionic network, compensated by thallium cations. Therefore, compound **3** cannot be treated as a double salt because there is no way to strictly assign the thallium cation (identification) to the nitrate anion moiety.

Attempts to compare the spatial molecular arrangement of compound **3** with known analogous systems containing potassium cations instead of thallium are not very meaningful. Despite the simplicity of the composition of the compounds obtained, it is surprisingly difficult to obtain crystals of a quality suitable for diffraction measurements. These unexpected difficulties have been known for a long time, and so far this problem has not been solved in an unambiguous way. The issue is well described in [53]. Additionally, the structure described in this publication contains Fe(III) and not, as in our case, Fe(II). Nevertheless, comparing the mode of interaction of K cations with cyanido nitrogen atoms with that involving thallium cations of structure **3**, one can come to a surprising conclusion that, despite the larger ionic radius, thallium cations interact more strongly with cyanido ligands (shorter distances) and also exhibit a much greater richness of interactions. This willingness to form many relatively strong interactions no doubt accounts for the high toxicity of thallium, which, when competing with potassium cations, simply binds more readily. These factors also contribute to the poorer solubility of thallium systems compared to potassium-containing compounds.

#### 3.1.5. Structures **4** and **5**

The structures of compounds **4** and **5** are isostructural. In both cases, the asymmetric part of the elemental cell contains 4 thallium cations, Tl(I) and one anion [W^IV^(CN)_8_]^4−^ (**4**) or [Mo^IV^(CN)_8_]^4−^ (**5**) (Figure 9). The anions have a slightly distorted square antiprismatic D_4d_ geometry, with average Me-C distances of 2.155 and 2.165 Å for W and Mo, respectively. The average C-N triple bond lengths in both structures do not deviate from the literature and are 1.153 Å. The angle values on the Me-C-N-cyanide bonds are nearly linear, with a deviation from 180° of less than 5 degrees, and are 176.6° for both metals. Both structures, **4** and **5**, show the presence of “side-on” and “end-on” interactions between thallium cations and cyanido ligands analogous to those in structures **1**, **2**, and **3**. Since **4** and **5** are isostructural, the example of structure **4** shows that the thallium cation, Tl1, interacts with three cyano ligands, of which two interactions are of the lateral type with C7N7 (Tl-N = 2.936 Å) and C6N6 (Tl-N = 2.862 Å), while one is of the “end-on” type with C5N5 (Tl-N = 2.878 Å). An analogous interaction pattern to Tl1 is exhibited by Tl2, where two “side-on” interactions (C4N4 and C7N7 with Tl-N = 3.010 Å and 3.004 Å, respectively) and one “end-on” interaction with the cyanide ligand C2N2 (Tl-N = 2.785 Å) are observed. The Tl3 cation shows three “end-on” interactions (C1N1, C8N8, and C3N3, Tl-N = 2.748–3.024 Å), whereas the thallium cation Tl4 is the only one of the cations to interact with four cyanido ligands, including three in the “side-on” and two in the “end-on” arrangement (Figure 10). Comparing the thallium structures of hexacyanides and octacyanides, less numerous interactions of cyanido ligands with thallium cation are observed in the latter. In the case of octacyanide systems, the number of interactions is more or less equally distributed between the metal cations (3–4 each), while in the case of hexacyanide systems, an uneven distribution of the number of interactions between the thallium cations can be observed. A lack of other molecules in the structure, e.g., water, nitrate anions, or bipyridyl, whose presence would cause the appearance of additional interactions, results in octacyanide systems, containing thallium cations, having an even spatial arrangement of the structure (packing); see Figure 11. It should be noted that for potassium analogues, interactions with cyanides are much poorer and less abundant, which results in much better solubility of them.

## 4. Conclusions

In the present work, the structures of five new thallium compounds were synthesized and discussed. In the studied compounds, the coordination of the central atoms in the anions was 8 for structure **5** (Mo) and structures **1**, **2**, and **4** (W) and 6 for structure **3** (Fe), while the coordination number of thallium was 2 (Tl3 in **3**), 3 (Tl4 in **3**; Tl1, Tl2, Tl3 in **4** and **5**), 4 (Tl2 in **3**; Tl4 in **4** and **5**), 5 (Tl1 in **1**; Tl1 in **2**) and 6 (Tl2 in **1**; Tl1 in **3**).

The similarity of the newly obtained thallium complexes to the already-known analogous potassium cation systems was discussed. The rare “side-on” bonding was shown for the first time and also for the first time in the case of the iron cyanido system. The “side-on” interactions of thallium in all cyanido complexes is an indication of the role of Prussian blue (and in general cyanido complexes) in the thallium detoxification procedure. Unlike insoluble Prussian blue, the K_2_[W(CN)_6_(bpy)] complex, which shows solubility in both water and organic solvents, appears to have better penetration properties in biological systems. It is also extremely inert and stable both in strong acidic and alkaline media. The pharmaceutical application of the described tungsten complex is now under intense investigation.

## Figures and Tables

**Figure 1 materials-15-04586-f001:**
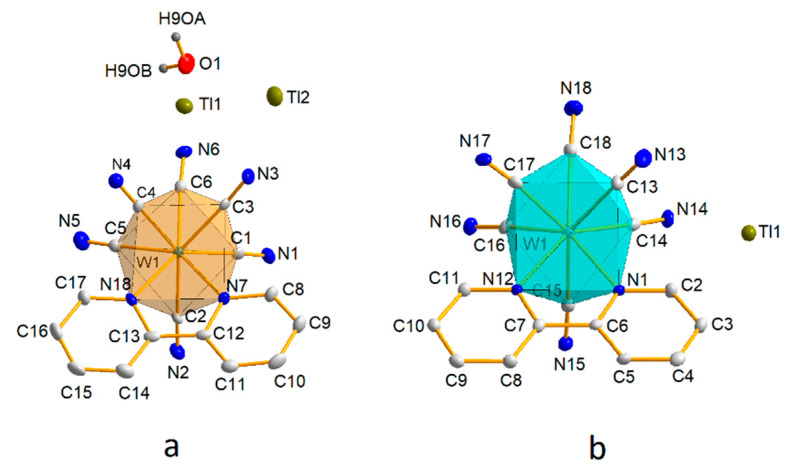
The asymmetric part of the cell unit of the compound **1** (**a**) and **2** (**b**) with the adopted numbering scheme. All hydrogen atoms of the bipyridyl ligand are omitted for clarity. All non-hydrogen atoms are represented at 30% probability thermal ellipsoids.

**Figure 2 materials-15-04586-f002:**
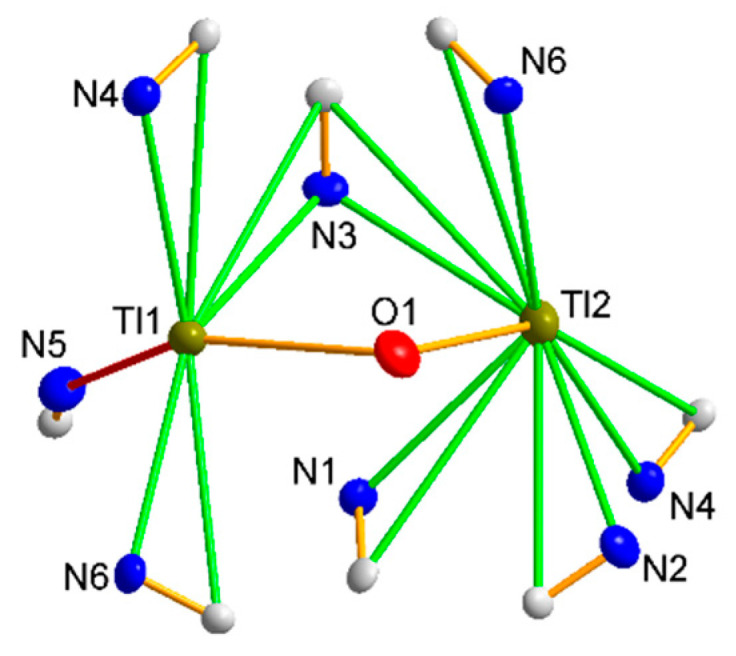
The coordination environment of two thallium cations. The “side-on” type bridging interactions are marked in green, and the “end-on” type in red. All H atoms are omitted for clarity. All atoms shown are represented at 30% probability thermal ellipsoids.

**Figure 3 materials-15-04586-f003:**
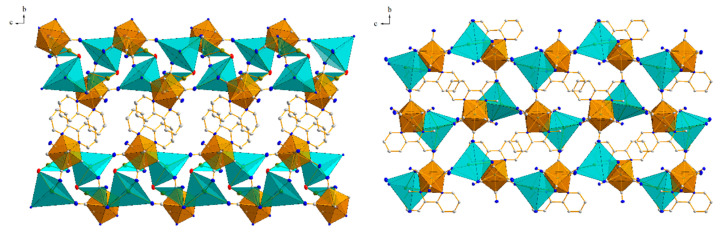
Packing diagrams in structure **1** (**left**) and **2** (**right**), both in projection in the (100) direction. The orange colour represents the coordination polyhedra of the tungsten W atoms, and the blue colour represents the thallium cations. Left: clearly visible layered packing character of structure **1**, also observed in all alkali analogues of [W^IV^(CN)_6_(bpy)]^2−^. Right: the characteristic 3D packing motif of structure **2** based on one [W^V^(CN)_6_(bpy)]^−^ anion connected by cyanide bridges to one Tl(I) cation. 30% ellipsoidal probability; hydrogen atoms are omitted for clarity.

**Figure 4 materials-15-04586-f004:**
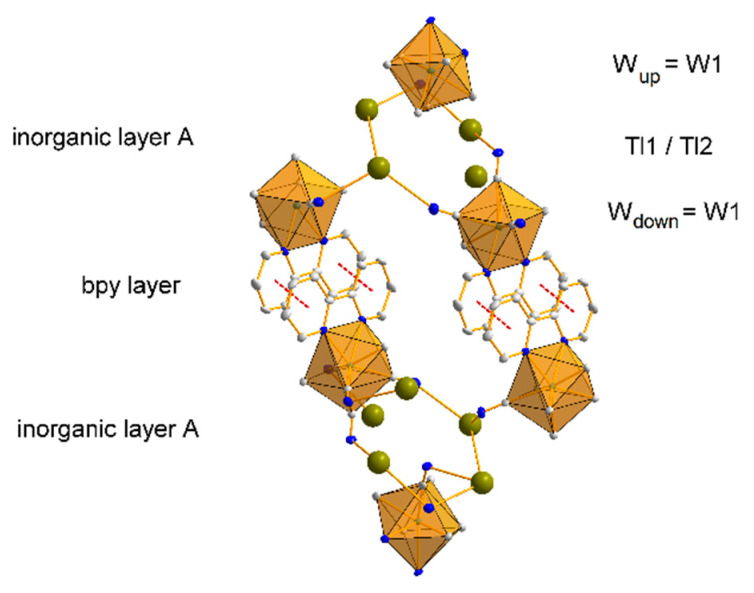
Fragment of structure **1** packing along the [010] direction with the π…π interactions (red, dashed lines) between the pyridyl rings of the bpy ligands marked. The figure contains a schematic of the layered structure of **1**, consisting of an inorganic layer A and a bpy layer. The inorganic layer A consists of three sandwiched sublayers, W/Tl/W. Some atoms have been removed for clarity.

**Figure 5 materials-15-04586-f005:**
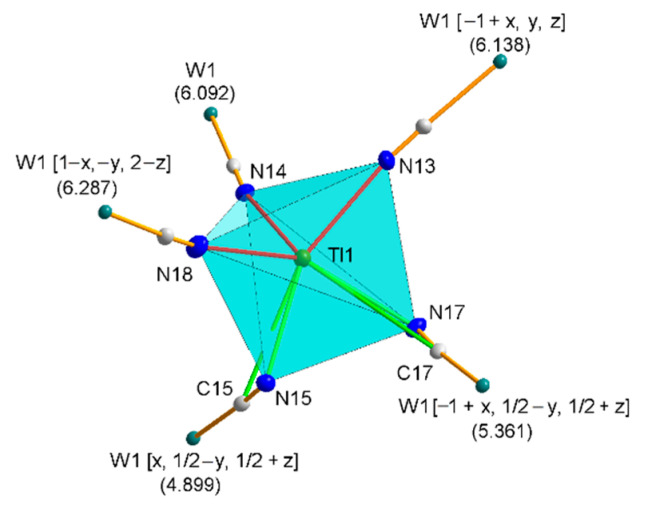
The geometry of the coordination environment of the thallium cation in compound **2**, along with the metallic environment. W-Tl distances in angstroms are given in parentheses. The “side-on” type bridging interactions are marked in green, and the “end-on” type in red.

**Figure 6 materials-15-04586-f006:**
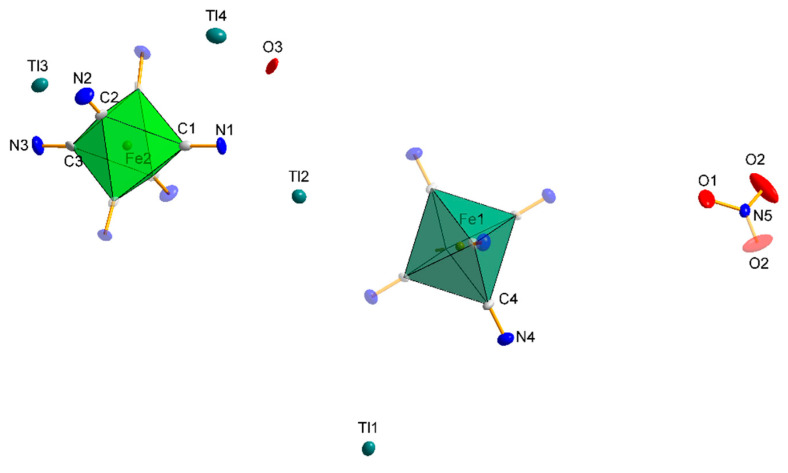
The asymmetric part of the cell unit of the compound **3** with the adopted numbering scheme. All atoms are represented at 30% probability thermal ellipsoids. Fragments of molecules duplicated by symmetry are shown in higher transparency.

**Figure 7 materials-15-04586-f007:**
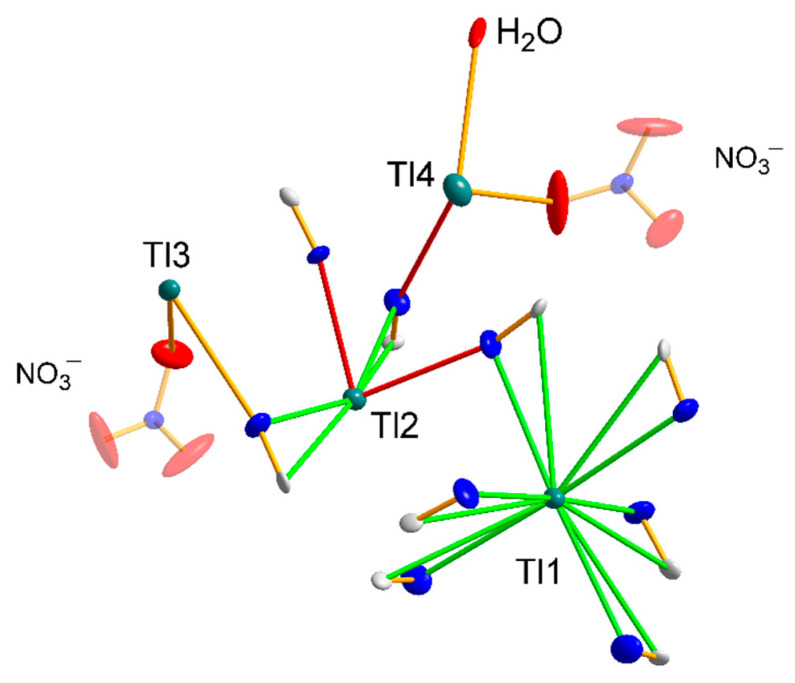
Coordination environment of the four thallium cations in structure **3** and their different bridging modes by cyanido ligands. “Side-on” bridging interactions are shown in green, and “end-on” interactions in red. All atoms shown are represented on thermal ellipsoids with a probability of 30%.

**Figure 8 materials-15-04586-f008:**
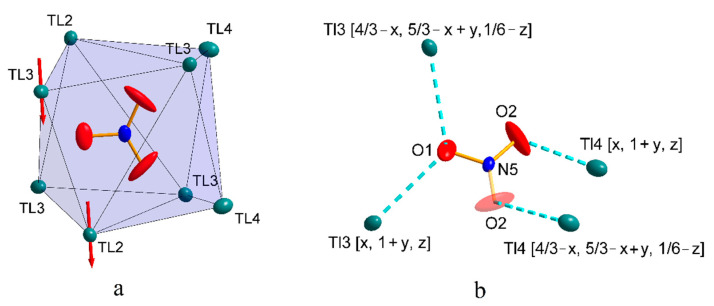
(**a**) Geometry of a slightly distorted antiprism square surrounding a nitrate anion formed by eight thallium cations. Red arrows indicate the direction of polyhedral deformation. (**b**) Strong nitrate anion interactions with thallium cations.

**Figure 9 materials-15-04586-f009:**
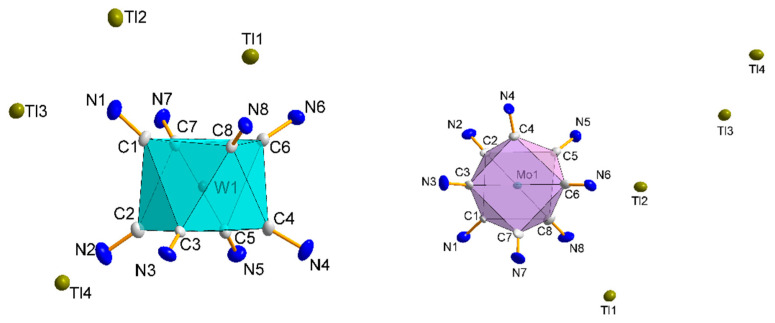
The asymmetric part of the cell unit of the compound **4** (**left**) and **5** (**right**) with the adopted numbering scheme. All atoms are represented at 30% probability thermal ellipsoids.

**Figure 10 materials-15-04586-f010:**
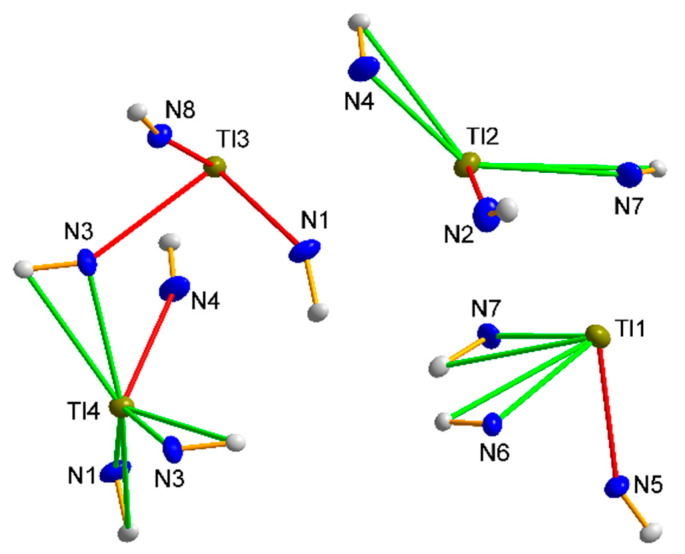
Coordination environment of the four thallium cations in structure **4** and their different bridging modes by cyanido ligands. “Side-on” bridging interactions are shown in green, and “end-on” interactions in red. All atoms shown are represented on thermal ellipsoids with a probability of 30%.

**Figure 11 materials-15-04586-f011:**
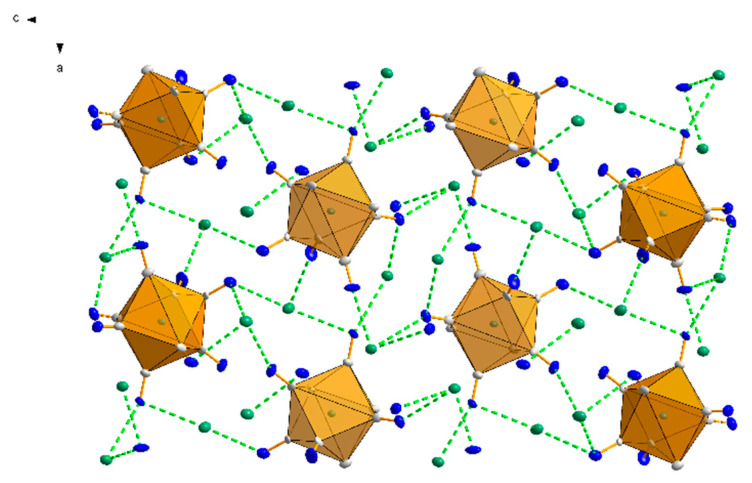
The spatial arrangement of molecules seen in the [010] direction in structure **5**. The green dashed lines indicate the interactions of thallium cations with cyanido nitrogen atoms.

**Table 1 materials-15-04586-t001:** π…π interactions for **1** and **2** [Å].

	π…π	Shift	
**1**			
Cg1…Cg2 [2 − X,−Y,1 − Z]	3.993(4)	1.840	Cg1: N7-C8-C9-C10-C11-C12, Cg2: N18-C13-C14-C15-C16-C17
Cg2…Cg1 [2 − X,−Y,1 − Z]	3.994(4)	2.097	
**2**			
Cg1…Cg1 [1 − X,−Y,1 − Z]	4.1066(1)	1.702	Cg1: N1-C2-C3-C4-C5-C6 Cg2: N12-C7-C8-C9-C10-C11
Cg1…Cg1 [2 − X,−Y,1 − Z]	4.1662(1)	2.362	

**Table 2 materials-15-04586-t002:** Hydrogen bonds for **1** and **2** [Å and °].

D-H…A	d(D-H)	d(H…A)	d(D…A)	<(DHA)
**1**				
C(16)-H(16)…N(2)#3	0.93	2.52	3.347(10)	147.8
C(11)-H(11)…N(4)#4	0.93	2.67	3.557(10)	160.8
C(17)-H(17)…N(5)	0.93	2.69	3.346(10)	128.7
C(14)-H(14)…N(4)#4	0.93	2.57	3.413(10)	151.0
O(1)-H(9OB)…N(3)#5	0.94(2)	1.97(4)	2.866(8)	160(7)
#1 x + 1/2,−y + 1/2,z − 1/2 #2 x − 1/2,−y + 1/2,z + 1/2 #3 −x + 2,−y,−z #4 −x + 2,−y,−z + 1 #5 x − 1/2,−y + 1/2,z − 1/2				
**2**				
C(5)-H(5)…N(17)#1	0.95	2.55	3.472(11)	164.3
C(8)-H(8)…N(17)#1	0.95	2.34	3.228(11)	155.2
C(2)-H(2)…N(16)#2	0.95	2.68	3.352(11)	128.5
#1 −x + 2,y + 1/2,−z + 3/2 #2 x,−y + 1/2,z + 1/2				

## Data Availability

Not applicable.

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
