# Peer review of "The Role of Prussian Blue-Thallium and Potassium Similarities and Differences in Crystal Structures of Selected Cyanido Complexes of W, Fe and Mo"

_materials, 2022, doi:10.3390/ma15134586_

Round 1

Reviewer 1 Report

The manuscript reported the structures of five new thallium complexes and their unusual high coordination numbers. In particular, the similarities (and the differences) with the well-known analogous potassium cation systems are reported. The higher strength of the bonding of thallium with respect to the potassium one may explain the role of Prussian blue in the thallium detoxification treatment. The study could be interesting for Materials readers, hence I suggest the publication after minor revisions.  

Minor revisions:

1) The title should try to highlight not only the similarities but also the differences between complexes.

2   2) Pg. 13, two sentences are repeated

3     3) In Conclusions Section, similar concepts are repeated.

4     4) In the Conclusion, the authors wrote that “The especially promising is the K2[W(CN)6(bpy)] complex, showing solubility both in water and in organic solvents thus having better penetration properties in biological systems”, but in the text there is not a comparison between the penetration properties in biological systems of the different complexes.

Reviewer 2 Report

The work by Hodorowicz et al is devoted to the studies of cyanocomplexes of W, Fe and Mo capable to interact with thallium ions to form complex compounds. The paper contains new original data siutable for publication,  The authors have obtained several novel compounds and studied it by the means of X-ray and have found the unusual side-on coordination of cyano ligands with Tl. The observed interaction between CN groups and Tl atoms allow to explain the use of cyanides for treatment of thallium ion poisoning. This fact makes the paper interesting and relevant for publication in Materials. The paper is well-written but is rather big which makes it not easy to read.

The main idea of the paper is to study the interaction of metal cyanides with thallium for its binding. Authors have shown that reaction of complex cyanides with thallium nitrate gives complexes 1-5. Unfortunately they provide no information about solubility of complexes 1-5 in water and other general solvents which is required to estimate its possible application in the mentioned fields. I think this information should be added to improve the quality of paper and its scientific significance.

The paper can be published in Materials after minor revision. To improve the paper the authors should:

1. Revise text provided at lines 300-345 to make it more short-spoken

2. Provide the information about the solubility of obtained complexes 1-5 in water and other general organic solvents.

Reviewer 3 Report

With more characteristic measurements of coordination complexes, the better approaching is provided in understanding colors and/or factors of metal complexes. Therefore it would be interested with some investigations of mixed heteronuclear complexes, which fit metal alloys as well as china goods. On the basis of this consideration, the paper is recommended but better with some revisions:

1  It would be necessary to adapt the Table 1, even it would be fine to move those information into supplementary information. In fact, authors could enphasize their findings of structural features. 

2  With my views, the figures would be somewhat reproduced according to any simarility of structural complexes, for example, compounds 1 and 2. In addition, the packing images should be also considered in saving spaces. 

3  The coordination numbers of individual metal ions should be better described. 
